# English as the Language for Academic Publication: on Equity, Disadvantage and 'Non-Nativeness' as a Red Herring

**Anna Kristina Hultgren** 

School of Languages and Applied Linguistics, The Open University, Walton Hall, Milton Keynes MK7 6AA, UK; kristina.hultgren@open.ac.uk

**Abstract:** Within the fields of English for Academic Purposes (EAP) and English for Research Publication Purposes (ERPP), the question of whether English as an Additional Language (EAL) scholars are disadvantaged by the pressure to publish in English continues to be debated. In this paper, I challenge this orthodoxy, raising questions about the evidence upon which it is based. Within a framework of 'verbal hygiene', I will argue that the attention accorded to 'non-nativeness' may be disproportionate to its significance for publication success. I conclude by proposing some reorientations for researchers and practitioners in the field that encompass non-linguistic structures of inequity.

**Keywords:** EAL disadvantage; non-nativeness as a red herring; verbal hygiene; field reorientations

## 1. Introduction

The ubiquity of English as the language of science and academia is indisputable. But to what extent are those who do not have English as their 'mother tongue' disadvantaged by this? This is a question that has preoccupied researchers and practitioners in EAP (English for Academic Purposes) and ERPP (English for Research Publication Purposes) for some time [1–4]. As Kuteeva and Mauranen put it: 'To date, most of the research has focused on the "centre" versus "periphery" dichotomy [ . . . ] and the challenges that non-anglophone researchers face when they try to publish their research in English-medium journals' [5]. Indeed, the acronym behind 'PRISEAL' (Publishing and Presenting Research Internationally: Issues for Speakers of English as an Additional Language), the conference which hosted many of the papers collected in this special issue, rests on an assumption that there are certain issues with publishing and presenting research internationally that are specific to speakers of English as an additional language.

Although the field has made a deliberate attempt to move away from a deficit view of non-native English research writers, signaled partly through a broadening of terms to refer to this group of individuals, such as 'EAL writers', 'non-anglophone scholars', 'second language research writers', 'plurilingual EALs' and 'multilingual scholars', the extent to which the field has truly managed to shed its deficit heritage can arguably be called into question. As Ken Hyland recently noted, the field is characterized by a 'pervasive view which asserts that EAL (English as an Additional Language) scholars are disadvantaged in the cut-throat competitive world of academic publishing by virtue of their status as second language writers' [2] (p. 10). The question of non-native English users' purported disadvantage has recently resurfaced more explicitly in Hyland's paper published in 2016 with the perhaps deliberately provocative title: Academic publishing and the myth of linguistic injustice, which prompted rebuttals and debates [2–4].

In the present paper, I revisit this important question, that is, whether non-native users of English are disadvantaged in the field of academic publishing. Although this is not the only issue of concern that has been raised in the context of the hegemony of English as an academic language—epistemicide being a case in point [6,7]—it will be the focus in this paper. However, rather than venturing a definitive answer to the question (and here is a spoiler alert: I do not think there is one!), I will approach it from a slightly different perspective. More specifically, I will analyze it—and the discourses that surround it—within a framework of 'verbal hygiene' [8,9]. Before I go on to explain what 'verbal hygiene' entails in more detail, it is worth noting that the debate about whether the increasing global dominance of English causes inequalities is of course not a new one. Nor is it unique to the specific fields of EAP or ERPP. In the wider field of Applied Linguistics, English has for several decades now been described variably as a Tyrannosaurus Rex, a Trojan Horse, a Hydra, a Cuckoo, a Killer Language and as causing Linguistic Imperialism and Linguicide [10–16]. So, while tapping into EAP and ERPP-specific debates about whether or not non-native users of English are disadvantaged, I will also question broader disciplinary axioms about the nature of disadvantage and its assumed linguistic basis.

First of all, however, I wish to make it abundantly clear that in questioning non-nativeness as consequential for academic publishing success, I am not denying that there *are* gross inequities and disadvantages in the field. Certainly, geolinguistic privilege has been convincingly documented by key scholars in the field [17–22]. Nor am I denying that the current system of global knowledge production and academic publishing is deeply flawed in many ways. Rather, I am going to raise the possibility that inequity, disadvantage and flaws exist not primarily in the linguistic realm, but elsewhere, and that conflating the issues risks misdiagnosing the problem and proposing the wrong solutions. The gross disadvantage and inequity that exists in global academic publishing is an indisputable fact [23]; however, as I will show, the overriding factor explaining this inequity appears to be, not whether or not English is a research writer's first language, but the resources and networks that are available to them. This reorientation away from language resonates with recent shifts in Applied Linguistics that seek to find ways of incorporating materiality into our theoretical and analytical frameworks [24–28]. What I will propose, then, is the—to some possibly controversial—idea that 'non-nativeness' may be a red herring. By 'red herring' I mean something that misleads or distracts from a relevant or important issue and that leads us towards a false conclusion. Of course, in doing so, care must be taken not to 'throw the baby out with the bath water' in terms of targeted support for multilingual scholars.

I begin by outlining my theoretical framework, that of verbal hygiene. I then go on to consider some of the discourses about EAL disadvantage that circulate, analyzing them within the framework of verbal hygiene. I proceed to calling for researchers to question their epistemological presuppositions, supporting this with evidence of non-Anglophone scholars' actual publication rates. I conclude by suggesting some extensions to current theories and practices in the field of EAP and ERPP that center on broadening the scope to account for non-linguistic structures of inequity alongside linguistic ones.

## 2. Verbal Hygiene

My argument is framed within the theory of 'verbal hygiene', as laid out by the British sociolinguist Deborah Cameron in her book with the same name, first published in 1995 and with a second edition in 2012 [8,9]. Central to the idea of verbal hygiene is that people have an irresistible urge to debate, discuss, and sometimes regulate and intervene in matters of language. To illustrate her point, Cameron draws on a wide range of examples of language-centered discourses and practices from recent British history. These include publishing houses who invent style guides to serve the interests of copyeditors who are paid to enforce them, 'political correctness' reformers who propose alternative words for the sake of 'fairness', 'sensitivity' and 'inclusivity', as well as members of the British Conservative Party who argued for reintroducing the teaching of grammar in schools as a symbolic way to uphold standards and reject the values of different social and ethnic groupings. Put in other words: social groupings, sometimes with widely differing ideological stances and vested interests, are predisposed to produce vast amounts of meta-discourse about language and sometimes try to police language use.

Verbal hygiene, then, refers to a 'motley collection of discourses and practices through which people attempt to "clean up" language and make its structure or its use conform more closely to their ideals of beauty, truth, efficiency, logic, correctness and civility' [8] (p. vii).

Another central point in the verbal hygiene framework is that language-related debates and interventions, such as those exemplified above, are rarely only about language. As Cameron herself puts it: 'complaints about language changes are usually symbolic expressions of anxieties about larger social changes' [8] (p. 238). So, for instance, when the older generation complain about young people using text speak, this can be seen as reflecting underlying anxieties about rapid technological developments and perhaps their own insecurities and self-perceived inadequacies in the area. Another illustrative example, which may resonate with readers across European nation states, is the conflation of xenophobia with matters of language. Writing about the case in Britain, Cameron suggests that when politicians 'bang[] on about the importance of English, and the menace of the immigrant who can't/won't speak it' [29], they are according disproportionate attention to matters of language and language competence. British media are known to have headlined scaremongering stories such as '22% of households in London contain no one who has English as their main language' and 'Polish now Britain's second language'. However, census data suggests that such stories are grossly exaggerated. Cameron cites the most recent census for evidence that only 1.6% of the population declare that they have limited or no proficiency in English, whereas those with no proficiency constitute less than 0.5%, a number that is likely to also include pre-school-aged children and people who have just arrived in Britain [29]. According to Cameron, this illustrates the tendency in contemporary society to make 'a mountain out of a mole-hill' where language is concerned, and that language often becomes the ground where political, social and moral debates play out. She explains:

> In any given time and place, the most salient forms of verbal hygiene will tend to be linked to other preoccupations which are not primarily linguistic, but are rather social, political and moral. The logic behind verbal hygiene depends on a common-sense analogy between the order of language and the larger social order, or the order of the world. The rules of language stand in for the rules that govern social or moral conduct and putting language to right becomes a sort of symbolic surrogate for putting the world to right. [30] (transcribed from an oral presentation)

If a preoccupation with language and language competences really is a cover preoccupation for some underlying anxieties and perceptions of how the world should be, then, as Cameron suggests, this would explain why opinions about language are often expressed with such passion and fervor. Because, as she says, in most cases they are 'not just debates about language' [30], but debates about the current state of the world and about how to put it right.

Cameron is, of course, not the only one who has pointed to the indexical nature of discourses about language. The work of the French sociologist, Pierre Bourdieu, is also to a large extent centered on a recognition that linguistic and other issues are often conflated; as he puts it: 'Linguistic struggles may not have obvious linguistic bases' [31] (p. 80). Scholars working broadly within the tradition of American linguistic anthropology have also, in their own way and from different perspectives, taken an interest in the discourses—or language ideologies, as they would refer to them—that permeate any society. What such scholars have in common is their endeavor to understand 'representations, whether explicit or implicit, that construe the intersection of language and human beings in a social world' [32] (p. 3). In other words, scholars of language ideologies are interested in what such ideologies index in terms of taken-for-granted understandings of social categories such as nation, gender, simplicity, beauty, authenticity, knowledge or any other category that is seen as important [32].

It has been suggested that in times of perceived destabilization of norms, circumstances in which we arguably find ourselves today given increased physical and virtual contact between language users, debates about language tend to intensify and norms become more explicitly negotiated [33,34]. This arguably applies to most societal domains, but no less so to that of global academic publishing,

a domain that is in the process of undergoing profound transformation. The implementation of bibliometric performance indicators and research evaluation regimes in higher education systems across the globe are among some of the changes that have contributed to recent decades' 'revolution' of higher education [22,35–38]. Possibly, it is in the context of such intense restructuring that norms come to be seen as being in need of overt negotiation, as captured in the emerging field of study 'language regulation in academia' [39]. It has been shown that there are an abundance of stakeholders involved in text production, what they refer to as 'literacy brokers' [40]. However, while discourses about language are likely to be as, if not more, rife in global academic publishing as in any other field, we must query, in the spirit of verbal hygiene, the basis of such discourses: what do they index?; what anxieties underpin them?; what evidence is there that any concerns are well justified?

What makes Cameron's 'verbal hygiene' framework stand out is that no one is exempt from it. Even us linguists, who normally pride ourselves on our descriptivist stance, are guilty of it. It is, according to Cameron, impossible for anyone to adhere to the ideal of descriptivism because rules about language are never neutral and never a given. Any choice one makes will bear certain values with it. As Cameron further argues, not only is descriptivism an unattainable ideal, it is also an undesirable ideal because it pretends that a descriptivism is value-free. Any ideology—including descriptivism—is value-laden. Instead of dismissing the practice of 'verbal hygiene' as a misguided and trite exercise, then, Cameron argues that discourses about language serve important functions for those engaged in it. In a similar vein, I will argue that discourses about language and EAL disadvantage serve important functions for those who engage in it. This is whether those who engage in it are practitioners in the field, such as journal authors, reviews, editors and EAP practitioners, or theoreticians, such as Applied Linguists in general or more specifically EAP and ERP scholars. I will also, in the spirit of 'verbal hygiene', reflect on whether the abundance of discourses about language and EAL disadvantage are proportionate to their effect in the 'real world'.

## 3. Discourses of EAL Disadvantage

As noted, the fields of ERP and EAP rest on the assumption of some form of 'native speaker advantage'. This assumption has been derived from empirical research of stakeholders engaged in the practice of academic publishing. By the 'stakeholders engaged in the practice of academic publishing', I understand everyone who, to varying degrees, has a stake in and is involved in the production and process of academic publishing. These stakeholders range from text producers themselves, most notably manuscript authors, to 'literacy brokers', such as 'editors, reviewers, academic peers, and English-speaking friends and colleagues, who mediate text production in a number of ways' [40] (p. 4).

One illustrative example among many of the discourses that prevail in global academic publishing are provided by Lillis and Curry (2015) [41]. In their study, the authors examined editors' and reviewers' comments on 95 manuscripts produced by 50 multilingual scholars from Slovakia, Hungary, Spain and Portugal working in the areas of psychology and education. They found that 58 of them (61%) foregrounded problems with the language or with English as a significant problem with the article. Of those 58 manuscripts where language or English is foregrounded as a problem, 47 (81%) are explicitly marked as related to English being used as a 'foreign language' or by 'second language users'. The editors' and reviewers' comments took the following form:

> The author is clearly well-versed in this area, but the manuscript needs quite a bit of work on its English grammar and spelling.

> The English language should be improved.

> The author(s) of the paper are clearly struggling with English as a second language, which is an issue in itself.

> The English has to be revised and worded in a more idiomatic and simple way.

> It is crucially important that the whole manuscript is proofread and edited by a native English speaker to make sure that all paragraphs convey the authors' intended meaning accurately.

Such comments clearly foreground these authors' status as second-language users, casting them as struggling and their language as in need of work, editing, proofreading, revision and improvement. As Lillis and Curry show, not only is language in general commented on extensively in reviews, the fact that the manuscript authors use English as their second-language is also topicalized. However, despite this topicalization of second-language users, Lillis and Curry are careful to point out that their data does not provide evidence that any negative comments on English actually lead to an article being rejected. As they say 'language by itself does not act as a warrant for dismissal or rejection' [41] (p. 147).

Where Lillis and Curry's study is among the few that explore the discourses of journal editors and reviewers, a larger proportion of research has explored the perceived difficulties experienced by EAL authors themselves. Such research tends to reveal a mixed picture with scholars at one and the same time appreciating the benefits of a shared language while also reporting on challenges with writing in a language that they do not know as well as their first language. They also sometimes share beliefs that they are being treated unfairly. In a study based on 10 qualitative semi-structured interviews with senior Spanish academics at the University of Zaragoza in Spain, half of whom were working in physical sciences and engineering and the other half in the social sciences, issues related to language and EAL disadvantage were certainly foregrounded [42,43]:

> you become unsatisfied because you don't write as easily as you would do it in Spanish. In Spanish, when you write you can express something in a way that you like, that looks nice, not a literary piece of art, but something that does not sound poor and this is what happens when using English, you don't say what you know, only what you can.

> (Scholar interviewed in [42])

Nevertheless, despite such sentiments, Ferguson et al. 2011 summarize their interview data as follows:

> The abiding impression left by the interview data, then, is that interviewees certainly did feel linguistically constrained in writing their papers in English, and this was felt to be burdensome. Clearly, writing in Spanish would have been easier, quicker and allowed more nuanced expression, particularly in the introduction sections, and in this sense interviewees were at some linguistic disadvantage relative to English native-speaker academics. That said, there is little evidence that the interviewees considered language constraints a barrier to publication, or even a major cause of the rejection of submissions. The latter depended more on the quality of the research than on the quality of the language. [43]

In other words, whilst these EAL researchers recognize feelings of injustice and inadequacy expressed by author manuscripts, they are also careful to point out that there is 'little evidence that the interviewees considered language constraints a barrier to publication, or even a major cause of the rejection of submissions'. This is arguably key and should lead us to be cautious about drawing too hasty conclusions about any expressed feelings of disadvantage and their relationship with publishing success.

Here, it is worth returning to an issue that was briefly discussed above, namely that whilst discourses about language are rife in any domain, they are perhaps particularly rife in the field of academic publishing. This is not only because, as discussed above, the field is under transformation; more to the point, perhaps, it is also a field centered on language work. A written text is the main, or only, thing an author is being assessed on and the most important thing that editorial gatekeepers base their decisions on. Everything rides, then, on this particular text, the manuscript, and any meta-texts that might accompany it, be it authors' cover letters or referees' reviews, etc. More broadly, academic publishing is a high-stakes, prestigious field with a lot of vested interest both for the individual and for

an institution. Participation in it can make or break an authors' career and give kudos to editors; the allocation of funding to many higher education institutions is contingent upon participation in the field; and publishing houses vie to participate in the lucrative industry of global academic publishing. Thus, it of little surprise if discourses about language are especially strongly felt and fervently expressed in academic publishing.

Notwithstanding this, this should not, in the spirit of verbal hygiene, stop us from probing into what these discourses about language might reveal in terms of underlying values, ideologies, morals, anxieties and epistemologies. For the journal editors and reviewers, the discourses might be interpreted as a way of regulating access to the field and preserving its prestige. In other words, those who submit to the journal need to follow certain language norms in order to have their manuscript accepted, as clearly the field would lose prestige if everyone was allowed access. Much like the publishing houses, exemplified by Cameron, who lay down arbitrary style guides to serve the interests of copyeditors who are then paid to enforce them, academic publishing could be seen as a field with vested interest to regulate, monitor and uphold standards to protect the privileges of those who successfully participate in it. For manuscript authors, the strongly held feelings they clearly express about their linguistic inadequacy and inferiority may not only be about language but also reflect a number of underlying anxieties that can be about a range of different things. To some, they may reflect anxieties about organizational and professional restructuring and the implementation of new and unfamiliar evaluation regimes; to others they may reflect views of research needing to be based on shared norms of 'understandability and clarity' [44]. Often, such norms will be portrayed as 'common sense', but in the verbal hygiene framework, common sense is not value-free. Pervading all of this may well be a shared ideology that language users must adhere to standard language norms [1] [45].

It is also worth considering whether the concerns expressed may be an artefact of the methodology used. Methodologies such as interviews and questionnaires are designed to elicit attitudinal data; data on what participants think about a particular topic. Such attitudinal methodologies are not necessarily factual accounts of an ontological reality. Rather, they are discourses reflecting and reproducing existing societal values and norms. As Denzin puts it, interviews are ' ... not a method of gathering information, but a vehicle for producing performance texts and performance ethnographies about self and society' [46] (p. 24). Participants' responses inevitably follow the researcher's agenda. And the researcher's agenda, in turn, is of course itself a product of discourses that circulate in society. Of course, this does not mean that interview- and survey-based research on this topic should be dismissed. This large body of work is hugely valuable in enhancing our understandings of scholars' attitudes, perceptions and/or lived experiences [18,19,21,36,42,43].

To sum up, while there is little doubt that feelings of hardship, inadequacy and injustice are genuinely felt by manuscript authors, and that matters of language and EAL status are frequently invoked and topicalized in manuscript reviews, it should not be automatically inferred that these concerns pan out in an actual failure to publish. Let us now consider the evidence for that more carefully.

### 3.1. Epistemic Reflexivity

In the framework of verbal hygiene, Applied Linguists and/or EAP/ERPP scholars should ask about the extent to which the strongly felt emotions about matters of language and EAL disadvantage are indeed consequential for publishing success. This, of course, does not mean that we should deny or ignore the clearly felt experiences of our research participants or brush them aside as Marxist 'false consciousness'. What it does mean, however, is that we need to engage in greater 'epistemic reflexivity' [47] in order to raise our critical awareness, query our own epistemological baggage, and reflect on the presuppositions in our field. To Salö

---

[1]    I am grateful to Maria Kuteeva for this point.

sociolinguistic research seems to end up showing and saying exactly what one would have expected it to show and say, based on the position—social, academic or otherwise—from which the research was produced. Often, this is because scholars embody the values of the group they investigate and, all too often, fail to create a rupture with their inherited view of the problem they investigate. [47] (p. 2)

Of course, such perpetuation of values does not always happen and it would be a mistake to dismiss research simply because it embodies a particular research perspective and/or ideologies. Nonetheless, Salö's point may mean not only that we scholars have a tendency to uncritically adopt the views of our participants, but we also design our research in a way that reinforces the disciplinary axioms in our field. In the particular case of EAP and ERPP, by looking for disadvantage, we find disadvantage. Many of us, of course, also occupy a dual role of both theoreticians and practitioners (as authors, reviewers or editors actively participating in the field), a dual role that might make it even harder to 'create' that 'rupture' Salö writes about.

In what he admits is a 'quick and dirty' count, Hyland compares Native English Speakers' (NES) and English as an Additional Language (EAL) scholars' publishing share [2]. His analysis is based on articles published in the top five journals by Impact Factor in six subject areas.

As can be seen, Table 1 shows that in 2011, EAL scholars had the larger part of the publication share: 56.7% compared to 43.3% for NES, although in 2000 NES had the larger share (61.2% compared to 38.8% for EAL scholars). There seems to have been a trend upwards since 2000, and certainly in recent years, the numbers suggest that EAL scholars may not fare as badly as one might have assumed. Of course, these numbers do not tell us anything about the trial and tribulations and feelings of disadvantage authors of both kind may have gone through to get their work published. The numbers only give us insights into number of manuscripts published, not how many have been submitted and rejected; drafted and never submitted; or indeed never drafted in the first place. The numbers also do not take into account the proportion of scholars who are EAL users. If EAL scholars constitute the larger proportion of the global academic workforce, then it is only right to expect that they should have a larger publishing share. There is also an additional complexity in that the analysis is based on a classification of scholars according to their name, which, as Hyland concedes, is a method fraught with difficulties. Nevertheless, on its own, the data does offer a useful complement to interview and questionnaire studies focusing on the challenges faced by non-Anglophone scholars. Reflecting on the sometimes brutal nature of reviewer comments, Hyland speculates:

It is possible that the frequency of these comments, and occasionally their bluntness, may lead EAL writers to believe that language has played a decisive role in the rejection of their contributions. [2] (p. 65)

This statement resonates with other researchers who conclude that despite the undeniable attention accorded to matters of language and non-nativeness by journal authors, reviewers and editors, there is little evidence that this is a decisive factor in the rejection or acceptance of a manuscript [41,42].

To sum up, it seems that when discourses on language and non-nativeness are complemented with data on actual publication success, the picture changes somewhat. It seems that perhaps some of these scholars' anxieties are just 'normal' verbal hygiene discourses, akin to the ones that circulate in any other societal domain. Of course, questioning the role of first language for publishing success is not to deny that there *are* gross inequalities in global academic publishing. Quite the contrary. The next section turns to this issue.

**Table 1.** First author for articles published in top five journals by Impact Factor [1].

|  | 2000 | | 2011 | |
|---|---|---|---|---|
|  | **NES** | **EAL** | **NES** | **EAL** |
| Biology | 424 (61.4%) | 267 (38.6%) | 740 (58.7%) | 521 (41.3%) |
| Elec Engineering | 214 (46.0%) | 251 (54.0%) | 256 (24.7%) | 780 (75.3%) |
| Physics | 109 (27.8%) | 283 (72.2%) | 714 (31.1%) | 1583 (68.9%) |
| Economics | 340 (79.4%) | 88 (20.6%) | 270 (68/5%) | 124 (31.5%) |
| Linguistics | 288 (74.8%) | 97 (25.2%) | 242 (61.2%) | 153 (38.8%) |
| Sociology | 312 (79.0%) | 83 (21.0%) | 284 (69.8%) | 123 (30.2%) |
| Overall | 1687 (61.2%) | 1069 (38.8%) | 2506 (43.3%) | 3284 (56.7%) |

[1] Source [2].

*3.2. Inequities in Global Academic Publishing*

Inequities in global knowledge production are abundantly clear. In a recent study using bibliometric data from Scimago's index of Elsevier's Scopus database, gross disparities are identified in the share of academic outputs [23]. Table 2 shows that only 10 countries in the world produce well over half of the world's total academic output (63.3%) with the remaining 221 countries producing the rest. In other words, the publication of academic output is concentrated in a small minority of countries. Moreover, and perhaps unsurprisingly, O'Neil demonstrates a statistically significant correlation between the GDP of a nation state and its share of scientific output (Table 3). In other words, the richer a nation, the more scientific output it produces.

Unlike Hyland's study [2], O'Neil does not take into account author's first language. As we know, global academia is a highly internationalized domain with many scholars working and living in countries where the dominant language is another than their first language. Also, although O'Neil's data does not explicitly show the language in which the output is published, it seems likely that it is predominantly if not exclusively written in English, given that indexed journals tend to be English-medium. Despite such complexities, however, both O'Neil's and Hyland's macro-level data on share of global publishing do seem important to consider alongside more micro-level self-reports of publication anxieties and frustrations. Certainly, these findings appear to cement the view that grotesque inequalities exist when it comes to participation in global knowledge production and that wealth is a key correlate for a nation's likelihood to participate.

**Table 2.** Comparison of scientific output and GDP [1].

| Countries | Documents | GDP |
|---|---|---|
| 1–10 | 63.3% | 66.0% |
| 11–20 | 17.1% | 12.4% |
| 21–30 | 7.6% | 5.6% |
| 31–40 | 5.1% | 4.4% |
| 41–50 | 3.0% | 3.6% |
| 51–60 | 1.5% | 2.3% |
| 61–70 | 0.8% | 1.6% |
| 71–80 | 0.5% | 1.1% |
| 81–90 | 0.4% | 0.6% |
| 91–100 | 0.2% | 0.6% |
| 101–231 | 0.5% | 21.6% |

[1] Adapted from [23].

If we look more closely at the nations that are in the top tier of global academic knowledge production, it becomes clear that the idea of EAL disadvantage is only partially supported. Certainly, the English-dominant countries Australia, United Kingdom, Canada and the United States feature in the top 10; but so too do the non-English-dominant countries Switzerland, Sweden, Netherlands, Germany, Spain and France. This suggests that the language of the author is not the only and may not

even be the most important factor determining publishing success, bearing in mind the aforementioned caveats that the data does not reveal authors' first language nor the language of publication. In other words, there are certainly inequities in global academic publishing, but they may be more economically than linguistically founded. To get your work published, it may be more important that you find yourself in the right environment and that you have access to the resources and networks you need, than whether or not you are a native speaker of English.

**Table 3.** Comparison of scientific output and GDP [1].

| Rank | Country | Docs/Population |
|:---:|:---:|:---:|
| 1 | Switzerland | 11.68 |
| 2 | Australia | 8.72 |
| 3 | Sweden | 8.61 |
| 4 | Netherlands | 7.34 |
| 5 | United Kingdom | 6.39 |
| 6 | Canada | 6.13 |
| 7 | Germany | 4.51 |
| 8 | United States | 4.25 |
| 9 | Spain | 3.96 |
| 10 | France | 3.77 |
| 11 | Italy | 3.75 |
| 12 | South Korea | 3.5 |
| 13 | Poland | 2.35 |
| 14 | Japan | 2.09 |
| 15 | Turkey | 1.19 |
| 16 | Iran | 1.16 |
| 17 | Russian Federation | 0.99 |
| 18 | China | 0.74 |
| 19 | Brazil | 0.72 |
| 20 | India | 0.24 |

[1] Adapted from [23].

Blommaert shares his views on the commodification on academic publishing and how this perpetuates global inequalities:

> Academics operate in an industrial model of production, in which their output [ . . . ] is increasingly, and continuously more extremely, commodified. [A]cademic publishers (some of which are among the world's most lucrative businesses) appropriate academic writings and put them behind paywalls. Such factors simply, and predictably, exclude the overwhelming majority of the world's scholars, especially those who do not belong to the privileged elites studying and working in generously funded institutions. These elites, I should add, can be found in Harvard, Cambridge, Berlin and Bologna, to be sure. But also in Delhi, Shanghai, Cape Town, Rio de Janeiro, Singapore, Cairo, Istanbul and Qatar. [48] (np)

In other words, there is no doubt that there are global inequities in academic publishing; however, according to Blommaert and others these are economically founded and commercially driven; they serve to create and exacerbate the power and wealth of the elites, irrespective of the languages spoken and the location of residence of these elites.

Thus, there is in all likelihood something else going on here that extends beyond the native/non-native speaker dichotomy and beyond language as the primary unit of analysis. Of course, EAP and ERPP scholars may rightly point out that by drawing attention to *linguistic* disadvantage, they are not thereby denying *economic* disadvantage. Certainly, this is clear in the use of constructs such as 'geopolitics' and 'center–periphery' alongside or instead of the linguistically based 'native/non-native' dichotomy (see [49] for an overview). Scholars could justifiably argue that they are using the term 'EAL

users' and related terms as a shorthand for all sorts of disadvantage, whether linguistic or economic. Nevertheless, I would argue that it is important not to conflate terms and that greater precision is called for. It is only by properly diagnosing the problem that we are able to devise proper solutions. If we misdiagnose, we end up prescribing the wrong medicine. The challenge we face, however, is how to do this without risking further disadvantaging an already disadvantaged population and making an inequitable situation worse.

## 4. Conclusions

In this paper, I have introduced the framework of verbal hygiene and argued that discourses about language are rife in any domain, not least in the high-stakes domain of global academic publishing. I have suggested, in line with the verbal hygiene framework, that discourses about language are 'normal' and often reflective of underlying values and ideologies. I have called for Applied Linguists to engage in epistemic reflexivity and consider the extent to which our disciplinary assumptions reproduce our research findings. If we are prepared to consider that the native/non-native dichotomy may be a red herring, then what do researchers and practitioners need to concern themselves with? I finish this conclusion by briefly outlining some possible extensions to current activities among researchers and practitioners in EAP and ERPP.

(1) **Systems in addition to individuals** The critical scholarship that underpins the paradigm should be welcomed and continued; however, we need to widen the scope to better understand the neoliberal systems and processes that undergird the current system of global knowledge production. Systems must be explored alongside individuals; material conditions need to be considered alongside linguistic ones. Bibliometric performance systems, research funding structures and evaluation regimes need to be scrutinized along with their role in producing and reinforcing global inequalities. This applies not only to researchers but also to practitioners, whose important work in providing support for scholars in the periphery could usefully encompass, to the extent that it does not already, broader career guidance on how to navigate, or subvert, the performance-based systems that are increasingly in place in universities worldwide.

(2) **Widening of methods** Commonly used interview and questionnaire studies are useful in helping us understand the woes and worries of international scholars; however, they could usefully be supplemented by a wider range of methodologies, including macro-level and ethnographic studies [2,19,23]. All methodologies have strengths and drawbacks, with the result that some things are obscured, and other things foregrounded. This means that the greater the range of methodologies employed to study a particular problem, the better an understanding we are likely to get.

(3) **Comparative studies** Where the focus remains on language, it would be useful to conduct comparative research. Comparing the views and practices of non-Anglophone scholars with Anglophone ones could reveal potentially useful findings about whether the frustrations so extensively documented among non-Anglophone scholars are shared by Anglophone scholars. There is evidence to suggest that they are, particularly among novice scholars [36,50–54]. For practitioners, this would mean attending, not only to EAL scholars, but all scholars who are in need and want of additional support.

The tendency in some Applied Linguistics corners to assume that injustice can be alleviated through linguistic intervention or by generating theories about linguistic disadvantage could be seen as missing the target. Certainly, it seems likely to only be able to scratch the surface of tackling the gross global inequities we are witnessing. This is because such inequities are socially, politically and economically, and not linguistically, founded.

**Funding:** This research received no external funding.

**Acknowledgments:** I fondly thank the organizers and delegates of PRISEAL 4 for fruitful discussions.

**Conflicts of Interest:** The author declares no conflict of interest.

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
