# Peer review of "English as the Language for Academic Publication: on Equity, Disadvantage and ‘Non-Nativeness’ as a Red Herring"

_publications, doi:10.3390/publications7020031_

Round 1

Reviewer 1 Report

This is an extremely interesting paper which makes a highly relevant contribution to our field. The use of the 'verbal hygiene' paradigm is both novel and useful. My only very minor concern is that the paper still bears some evidence of originally having been a plenary lecture e.g. line 42 page 2. 

I would also like to see reference to this special issue of the journal as 'proceedings' (line 30, page 1) taken out. In fact, the issue will include papers authored by researchers who did not take part in the PRISEAL conference. It is likely to be problematic for others publishing in the issue if their paper comes to be considered (e.g. by research evaluation agencies) as a paper delivered at a conference and then published as is in conference proceedings. In Spain at least papers in proceedings are of much lower status than research articles published in journals. 

Author Response

Thank you very much for the insightful comments. Please see my detailed responses attached.

Reviewer 2 Report

The piece is provocative and well worth publishing. The author's position is well argued but not unproblematic - I have outlined my main concerns below - and a tempering of claims and inclusion of countering opinions is necessary prior to publication, in my estimation. Further minor comments are embedded within the attached PDF of the piece.

Concerns

1. Dismissiveness: Readers - particularly those who have contributed so much to the burgeoning field thus far through their qualitative research - may feel that you are too hastily dismissing their contributions. I suggest making minor changes here and there to address this concern (inclusion of more references; suggestion of the importance of the past decades of qualitative, often identity-focused research in the field of ERPP).

2. Strength of claim(s): Beware of making claims that are not supported by a wide body of research. Bibliometric data can be useful, but, as you point out, does not provide solid footing from which to forward some claims. A simple tempering of claims here and there would help.

3. Lack of reference to the implications for practitioners: I would have liked to see some implications for adopting your stance for those of us whose task it is to support plurilingual EAL scholars writing from the peripheries. Please include at least a sentence or two on this topic in the conclusion - again, a nod to those who undertake this important work (regardless of which "camp" they fall in) would be nice to see and make your argument easier to digest (not that it always has to be...I sure like to rock the boat to, when possible). 

Otherwise, I am impressed with the piece. It is not only well-written but also thought-provoking for those of us who wear several hats (authors; researchers; practitioners). I look forward to seeing the piece in print in the near future!

Author Response

(The authors gave the same response as above.)
